# OpenReview forum: "Memory Makes The Poison: Over Memorization Drives Visual Poisoning in LVLMs"
_ICLR.cc/2026/Conference — ICLR 2026 Conference Withdrawn Submission_

### Official Review · Reviewer_9DqD · 2025-10-26

**Soundness:** 1
**Presentation:** 1
**Contribution:** 1
**Rating:** 0
**Confidence:** 4

**Summary:**

The paper claims to find that memorization is the root cause of the success of data poisoning attacks in multimodal LLMs and conducts extensive experiments to verify their claim. And then propose RejectShield which filters out any data with poison and finetunes the model without such data, ultimately finding that such a process results in much lower ASR therefore a successful cleaning approach.

**Strengths:**

The study of data poisoning attacks in LLMs and multimodal LLMs is a very pertinent subject of study.

**Weaknesses:**

In the current state, the paper has several and serious flaws:
1. First of all it is commonly understood that memorization is the cause of data poisoning, otherwise such low percentages of poisoning will never lead to model being poisoned so effectively as demonstrated in this and several previous works. Therefore it is not the contribution of this paper to claim that memorization leads to data poisoning, it is kind of obvious.
2. The claims about multimodal being more subject to data poisoning is also something previous papers have found, see https://arxiv.org/abs/2106.09667.
3. The approach RejectShield which should have been the main contribution of this paper if it is such an effective technique has been treated as second class citizen in the work, there is hardly any mention of how it works. The authors just say that they have built a technique that can detect if an image has adversarial perturbation. As far as my understanding of adversarial attacks in images goes, this is an unsolved problem in the image community since 2013 when adversarial attacks in images were first discovered and there exist no reliable classifiers that given an image can predict if the image has an adversarial attack or not. So I am very curious of the details of how the authors of this work were able to achieve that. Please provide more details of the training data, training procedure, and the robustness of this classifier.
4. There are several possible errors in the paper. For example in line 180 (x_d, y_d) is not the benign image, it is the target pair, right?
5. Another mistake is in line 211 where the authors state the model was trained on the benign counterparts x_d (and not x_p), if that is the case then how did the model learn any poison (like where did the 1% poison in Figure 2 come from)? Can you explain this in much more clarity in the paper?

**Questions:**

Please refer to the weakness section.

---

### Official Review · Reviewer_dCKW · 2025-10-28

**Soundness:** 2
**Presentation:** 2
**Contribution:** 2
**Rating:** 4
**Confidence:** 4

**Summary:**

This paper finds that visual data poisoning in LVLMs is mainly caused by over-memorization during fine-tuning, not visual perturbations. The authors show that LVLMs hallucinate even when trained on benign but repetitive samples and that multimodal inputs worsen this effect. They reinterpret the ShadowCast attack as a memorization issue and propose RejectShield, a rejection-based defense that cuts attack success by up to 99% while preserving model performance.

**Strengths:**

1. The structure of the paper is clear, and the problem is well motivated.
2. The authors demonstrate the problems of the Shadowncast, which makes sense to me.

**Weaknesses:**

Main Concerns
1. This paper shows that the effectiveness of Shadowcast does not come from the poisoned attack, but the hallucination of the model. I think this makes sense to me, as the injected images contain only a single class, which likely introduces strong data bias and induces hallucination. However, it remains unclear whether this limitation is unique to Shadowcast or shared across other poisoning setups. To support the claim that over-memorization, rather than poisoning, drives the observed effect, the authors should conduct additional experiments under varied training configurations and data distributions. Without such evidence, the generality of the conclusion remains uncertain.
2. The paper only evaluates the Shadowcast attack. Including more advanced attack methods (e.g, [1]) could strengthen the robustness of the conclusion.

Other Concerns
1. The authors only evaluate the LLaVA v1.5 and MiniGPT-4 which are a little bit outdated right now. It is suggested that the authors introduced one more advanced VLM (e.g., Qwen2.5-VL-7B or Qwen3-VL-7B).

**Questions:**

1. How did the authors train the classifier? If the training data only consists of injected pairs (w/o perturbation) and clean data, how would this classifier perform? Since underlyingly, it is a hallucination problem, using a classifier to detect the adversarial examples might not be ideal solution.

---

### Official Review · Reviewer_9Fb1 · 2025-10-31

**Soundness:** 1
**Presentation:** 1
**Contribution:** 1
**Rating:** 0
**Confidence:** 4

**Summary:**

This paper argues that the true cause of data poisoning in LVLMs is the memorization of fine-tuning concepts, not the pixel-level perturbations themselves. It also introduces RejectShield, a simple rejection-based defense that filters out likely poisoned samples using an adversarial detector.

**Strengths:**

The problem setting is interesting, backdoor attacks are becoming an increasingly pertinent threat as more and more SFT data is mined from uncurated web sources.

**Weaknesses:**

## Weaknesses
- **Overstated Claims** The observation that backdoor attacks are successful due to models memorizing patterns is obvious and this paper is certainly not the first one to observe this. The authors repeatedly state that "backdoors are not successful because of the pixels" but what does this even mean?
- **Unsurprising Result in Figure 1** This result is unsurprising, as the pixel perturbations in backdoor attacks are meant to (1) enforce imperceptibility and (2) to prevent interference with other concepts that may exist in the finetuning dataset.  In the setting you have selected, the effect is likely observed due to their choice of concepts which are rare, low-overlap categories like engine lights naturally face little interference from clean data, making memorization-driven attacks appear stronger than they would for more common concepts. The broader implication of this plot is not at all clear to me.
- **Method not clearly described** Most of the paper is spent explaining a fairly obvious finding, and very little time is spent on describing RejectShield in the main paper though this is the main method of this paper. As far as I understood, it is a classifier that is able to detect adversarial examples but almost no details are provided on the datasets used to train this classifier or its error patterns, nor is there any sensitivity analysis.

**Questions:**

See weaknesses

---

### Official Review · Reviewer_zsrd · 2025-11-01

**Soundness:** 2
**Presentation:** 3
**Contribution:** 2
**Rating:** 4
**Confidence:** 4

**Summary:**

The paper studies how visual perturbation–based data poisoning attack, specifically ShadowCast, poses threats to Large Vision–Language Models, and the reasons existing input purification defenses appear to not fully mitigate the attack. The paper identifies the issue as the LVLM over memorizing concepts contained by the images added to the clean dataset, instead of the adversarial perturbations, by fine tuning with samples with only benign images without edits. Motivated by the observation, the paper proposes to filter the fine tuning dataset with an adversarial sample detector, complemented by an LLM that checks if there is imbalanced data in the dataset.

**Strengths:**

1. The paper conducts detailed analysis through controlled experiment to fine tune with only benign target samples, to motivate the proposed defense.

2. The paper conducts similar controlled experiments on single modality model to identify the problem is more serious for LVLM.

**Weaknesses:**

1. In Figure 1, labeling in the rate where the model mistakenly produces results with respect to x_d instead of x_o for the controlled experiment is a bit misleading, since the misidentification is not the intended outcome.

2. The experiment results suggest the reason the ShadowCast attack appears to be not mitigated by just input purification defenses is that the target model cannot distinguish samples containing the destination concept and samples containing the origin concept, due to that the model is fine-tuned with an unbalanced amount of target samples. I think there are some open questions the paper does not address that make it unclear about how to interpret the results of the paper. For example, if under the controlled experiment or after input purification, the target model does not only respond the prompts about the origin concept with information about the destination, but also respond to prompts about some other similar concepts with information about the destination concept, then it suggests what is observed is hallucination as a side effect of the targeted attack, and the targeting itself has been mitigated. Then the proposed defenses are closer to methods that mitigate hallucination when there is an imbalance of fine tuning data, instead of defenses for targeted attacked. In particular, using an adversarial perturbation detector works because the attacker has label the imbalanced data with adversarial perturbation.

**Questions:**

Q1.
> This vulnerability is particularly concerning, as it shows that standard LVLM fine-tuning can be destabilized with only a few injected samples.

Can the authors quantify "a few"? What is the typical fine tuning dataset size? For example, if the size if 10K, then 100 poison samples is not a trivial number.

Q2. If the origin concept and the target concept are more distinct under certain perception, like Harry Styles vs Biden, does the conclusion from Section 3.2 still hold?

Q3. If the there are benign samples containing origin concept and the target concept (some Trump example, some Biden example), does the conclusion from Section 3.2 still hold?

Q4. In the case study with engine light and fuel light, and only normal fuel lights examples are provided in fine tuning, if the fine tuned model is tested with an origin concept is some other warning light on a car that looks similar, would the model also output fuel light related answers?

Q5. Does the defense only improve on attack cases where origin concept images look similar to target concept image?

Q6. How does “ShadowCast" compare with the "controlled experiment" in regime where origin concept images are distinct from target concept image? Can “ShadowCast still generalize to other models under that regime.

---

### Note · Authors · 2025-11-14

I have read and agree with the venue's withdrawal policy on behalf of myself and my co-authors.